# *Pseudomonas simiae* WCS417 and *Debaryomyces hansenii* Induce Iron Deficiency Responses in Rice (*Oryza sativa* L.) Through Phytosiderophore Production and Gene Expression Modulation

**DOI:** 10.3390/plants14243769

**Published:** 2025-12-10

**Authors:** Jorge Núñez-Cano, Francisco J. Ruiz-Castilla, Francisco J. Romera, José Ramos, Carlos Lucena

**Affiliations:** 1Departamento de Agronomía (Unit of Excellence ‘María de Maeztu’ 2020-24), Edificio Celestino Mutis (C-4), Campus de Excelencia Internacional Agroalimentario de Rabanales (ceiA3), Universidad de Córdoba, 14071 Córdoba, Spain; jorgenunezcano@gmail.com (J.N.-C.); ag1roruf@uco.es (F.J.R.); 2Departamento de Química Agrícola, Edafología y Microbiología, Edificio Severo Ochoa (C-6), Campus de Excelencia Internacional Agroalimentario de Rabanales (ceiA3), Universidad de Córdoba, 14071 Córdoba, Spain; fjruizcastilla@hotmail.com (F.J.R.-C.); mi1raruj@uco.es (J.R.); 3Departamento de Botánica, Ecología y Fisiología Vegetal, Edificio Celestino Mutis (C-4), Campus de Excelencia Internacional Agroalimentario de Rabanales (ceiA3), Universidad de Córdoba, 14071 Córdoba, Spain

**Keywords:** *Oryza sativa*, iron deficiency, phytosiderophores, *Pseudomonas simiae*, *Debaryomyces hansenii*, gene expression, ethylene signaling

## Abstract

Iron (Fe) is an essential micronutrient for crop productivity, but its low availability in alkaline and calcareous soils limits the growth of rice (*Oryza sativa* L.), which employs a combined strategy for its acquisition based on the release of phytosiderophores (PS) and the use of specific transporters. In this study, the effect of the rhizospheric bacterium *Pseudomonas simiae* WCS417 and the halotolerant yeast *Debaryomyces hansenii* CBS767 as inducers of responses to Fe deficiency in rice grown under hydroponic conditions was evaluated. Plants were inoculated in nutrient solutions with and without Fe, and PS production and the expression of genes associated with biosynthesis and transport were determined by qRT-PCR. The results showed that both microorganisms significantly increased PS production compared to controls, especially under Fe-deficient conditions, although *P. simiae* also exerted an effect under Fe sufficiency. Furthermore, induction of key genes (*OsNAAT*, *OsIRO2*, *OsTOM1*, *OsYSL15*, and *OsIRT1*), as well as genes related to the ethylene pathway (*OsEIN2*, *OsACS2*, and *OsACO3*), was observed, pointing to a regulatory role for this hormone in the response. In conclusion, *P. simiae* and *D. hansenii* act as inducers of Fe acquisition mechanisms in rice, offering a sustainable biotechnological approach to improve iron nutrition in limiting environments.

## 1. Introduction

Iron (Fe), although required in small amounts compared with macronutrients such as nitrogen, phosphorus, or potassium, plays a fundamental role in plant health and productivity. Without adequate Fe, plants cannot perform essential functions, resulting in reduced growth and yield. This element is mainly concentrated in chloroplasts and mitochondria, where it participates in the photosynthetic and respiratory electron transport chains, as well as in chlorophyll biosynthesis [1,2]. Roots are responsible for Fe uptake from the soil or nutrient solution, and they have mechanisms to improve Fe solubility and absorption, especially under Fe deficiency. For example, roots can exude protons, organic acids, and phytosiderophores to solubilize Fe and enhance its acquisition [3].

Despite being the fourth most abundant element in the Earth’s crust, Fe availability in soil is often limited due to its tendency to form insoluble compounds, particularly under alkaline conditions [3]. To overcome this limitation, plants have evolved specialized strategies to improve Fe solubility and uptake [3]. This paradox has forced plants to evolve complex mechanisms to improve Fe solubility and optimize its uptake. The first evidence of these mechanisms came with the discovery of the ability of plants to acidify the medium [4] and to reduce Fe [5] under Fe-deficient conditions. Later, the observation of differences between graminaceous and non-graminaceous species led to the hypothesis that distinct mechanisms exist for Fe acquisition. This hypothesis was confirmed by Takagi [6] with the discovery of Fe-chelating compounds, known as phytosiderophores (PS), released by grasses. Today, two strategies are recognized for Fe acquisition in plants: Strategy I and Strategy II [7,8,9]. These strategies are not only evolutionary adaptations but also have profound implications for agriculture, since understanding them can pave the way for improving crop yield and nutritional quality in Fe-deficient soils [10].

Rice employs a combined strategy, as it can use mechanisms from both Strategy I and II. Flooded soils initially generate Fe^2+^ availability, but prolonged flooding and pH changes lead to insoluble forms, causing Fe deficiency. Rice plants exude specific PS such as 2′-deoxymugineic acid (DMA), synthesized through a pathway involving S-adenosylmethionine (SAM), nicotianamine (NA), nicotianamine aminotransferase (NAAT), and deoxymugineic acid synthase (DMAS) [11,12,13,14,15]. DMA secretion is mediated by the TOM1 transporter [16], and uptake of the Fe^3+^/DMA complex is facilitated by *OsYSL15* and *OsYSL16* [17,18,19]. In addition, rice expresses Fe^2+^ transporters such as *OsIRT1* and *OsIRT2* [20,21]. Among the transcription factors that regulate these responses, *OsIRO2* plays a central role as an iron-related basic helix–loop–helix (bHLH) regulator that positively controls the expression of PS biosynthetic and transporter genes, including *OsNAAT1*, *OsTOM1*, and *OsYSL15*. In contrast, *OsIRO3* and *OsbHLH133* act as negative regulators of Fe-deficiency signaling [22,23,24].

Given the agronomic importance of rice and the challenges posed by Fe deficiency, the role of beneficial microorganisms in modulating Fe acquisition mechanisms has received increasing attention. Rhizospheric bacteria and yeasts can influence Fe solubility through siderophore production, acidification, or stimulation of plant responses [25,26]. Among bacteria, *Pseudomonas simiae* WCS417 is a well-studied strain known for its ability to enhance nutrient uptake and induce systemic responses in host plants [27,28,29]. However, most studies have been conducted in soil systems, without directly addressing its effect on PS production in rice. On the other hand, the non-conventional yeast *Debaryomyces hansenii* has been described as a halotolerant microorganism with plant growth-promoting potential. It has been shown to enhance Fe uptake in dicotyledonous Strategy I plants [30] and to improve nutritional status in rice [31]. Nevertheless, its ability to trigger Fe-deficiency responses in rice, particularly PS synthesis and related gene expression, remains unexplored.

In this context, evaluating the effect of *P. simiae* and *D. hansenii* on Fe-deficiency responses in rice offers novel insights into their potential as biofertilizers. Specifically, their influence on PS production, the expression of biosynthetic and transporter genes (*OsNAAT*, *OsIRO2*, *OsTOM1*, *OsYSL15*, *OsIRT1*), and the possible involvement of ethylene as a regulatory signal is of particular interest. Ethylene has been widely implicated in the regulation of Fe-deficiency responses in Strategy I plants [32,33], and its interaction with microbial signals may play a similar role in rice.

The present study therefore aims to assess the role of *P. simiae* WCS417 and *D. hansenii* CBS767 as inducers of Fe-deficiency responses in rice plants grown under hydroponic conditions. By integrating physiological, biochemical, and molecular analyses, this work provides evidence for the capacity of these microorganisms to stimulate PS production and modulate gene expression, highlighting their potential for sustainable strategies to improve Fe nutrition in rice cultivation.

## 2. Results

### 2.1. Phytosiderophore Production in Rice Plants Inoculated with Pseudomonas Simiae or Debaryomyces Hansenii

Measurements of PS released by the roots of rice plants inoculated with the WCS417 strain of *P. simiae* were made at 24, 48, 72 and 96 h (Figure 1).

Overall, inoculation with the rhizospheric bacterium *P. simiae* (strain WCS417) significantly induced PS production in rice plants under both Fe sufficiency and deficiency conditions. In the first sampling carried out at 24 h, inoculated plants, both in the presence and absence of Fe in the nutrient solution, already showed significant differences with respect to their non-inoculated control. This behavior was similar in the 48 and 72 h samplings. At 96 h, PS production was higher and with highly significant differences in all inoculated treatments versus their non-inoculated control.

In the same way as what was done with *P. simiae*, it was done with the yeast *D. han-senii*, the measurements of PS released by the roots of the rice plants inoculated with the CBS767 strain being made at 24, 48, 72 and 96 h (Figure 2).

The effect of *D. hansenii* on PS production in rice plants varied depending on the Fe deficiency or sufficiency treatment in which the plants were placed (Figure 2). In plants grown with Fe sufficiency, inoculation with the yeast resulted in a decrease in PS production compared to control plants at virtually all sampling times. However, in the opposite case, where plants grew under Fe deficiency, statistically significant increases in PS production were observed compared to the control, caused by the presence of the yeast *D. hansenii* in the samples taken at 72, 96, and 120 h after inoculation.

### 2.2. Effect of Inoculation with the Bacterium P. simiae (WCS417) or the Yeast D. hansenii (CBS767) on Gene Expression Levels Related to Responses to Iron Deficiency

The expression of *OsNAAT1*, *OsIRO2*, *OsTOM1*, *OsYSL15* and *OsIRT1* was analyzed to evaluate the effect of microbial inoculation on genes associated with the phytosiderophore-mediated Fe uptake strategy. *OsNAAT1* encodes nicotianamine aminotransferase, an enzyme catalyzing a key step in the biosynthesis of 2′-deoxymugineic acid (DMA). *OsIRO2* encodes an iron-related basic helix-loop-helix (bHLH) transcription factor that acts as a positive regulator of DMA biosynthetic and transport genes, including *OsNAAT1*, *OsTOM1* and *OsYSL15* [22,23]. *OsTOM1* encodes the transporter responsible for DMA secretion to the rhizosphere, *OsYSL15* encodes the transporter mediating Fe(III)–DMA uptake, and *OsIRT1* encodes an Fe(II) transporter contributing to Fe acquisition under Fe-deficient conditions [3].

#### 2.2.1. Effect of the Bacterium *P. simiae* (WCS417) on the Expression of the Genes *OsNAAT*, *OsIRO2*, *OsTOM1*, *OsYSL15*, and *OsIRT1*

In the case of the *OsNAAT* gene, a highly significant induction of its expression was observed in plants grown under Fe-sufficient conditions compared to control plants at 24 and 96 h (Figure 3a). In turn, under Fe-deficient conditions, the expression of this gene involved in PS production was statistically significant compared to the levels found in control plants at all sampling periods (Figure 3b). Furthermore, expression was much higher than in plants grown without Fe but inoculated with this microorganism (Figure 3a,b). In the case of *OsIRO2* expression, the increase in expression levels under Fe-sufficient conditions was progressive until reaching its highest level of expression at 72 h and then decreasing again to basal levels (Figure 3c). Under Fe deficiency conditions, the highest levels of gene expression with respect to the control were reached in plants inoculated with the bacteria from 72 h (Figure 3d), decreasing again in the 96 h sampling but remaining significantly higher than the basal levels, which did not occur under Fe deficiency conditions (Figure 3c).

Regarding the effect of the bacteria on the expression of genes encoding Fe transporters, we found that there are highly significant differences in the expression levels of the transporter gene *OsTOM1* at 24, 48 and 72 h (Figure 4a) in rice plants inoculated with the bacteria and grown under Fe sufficiency conditions. The same occurs in all sampling periods under Fe deficiency conditions, in relation to non-inoculated plants, although with more variable expression levels between samples (Figure 4b). For the *OsYSL15* gene, it was observed that in the samples taken at 48 and 72 h, there were expression levels that differed significantly from the control under Fe sufficiency conditions, but at 96 h the expression returned to basal levels (Figure 4c). However, when the plants grew without Fe in the nutrient solution, inoculation under these conditions caused this gene to present high levels of expression in inoculated plants and with significant differences with respect to the control after 48 h of sampling (Figure 4d). Finally, *OsIRT1* showed greater induction of its expression in plants inoculated with the bacteria compared to the control plants only at 96 h under Fe sufficiency conditions (Figure 4e), while in Fe deficiency and inoculated with the bacteria, the activation of this gene began at 24 h and then decreased at 72 and 96 h (Figure 4f).

#### 2.2.2. Effect of the Yeast *D. hansenii* (CBS767) on the Expression of the Genes *OsNAAT*, *OsIRO2*, *OsTOM1*, *OsYSL15*, and *OsIRT1*

The involvement of this yeast in the expression levels of genes directly involved in PS production under Fe-deficient conditions in rice plants was evaluated at 24, 48, 72, 96 and 120 h and is shown in Figure 5. The OsNAAT gene showed a progressive induction in its expression compared to the control, reaching its maximum at 96 h in inoculated plants grown under Fe-sufficient conditions and at 72 h under Fe-deficient conditions, its evolution being somewhat more erratic than in the previous case (Figure 5a,b). Regarding the expression of the OsIRO2 gene, under Fe-sufficient conditions, inoculation caused a decrease in expression during the first samplings until it returned to basal levels in the last one. On the other hand, in plants with iron deficiency and inoculated with this yeast, a high transcriptional activation of this gene was found 24 h after inoculation, after which it decreased to basal conditions (Figure 5c,d).

Expression levels for the Fe transporter gene *OsTOM1* were significantly higher than the control from 72 and 96 h under Fe-sufficient conditions when inoculated with this yeast (Figure 6a), while under iron-deficient conditions, the same inoculation with the yeast showed an induction of transcriptional expression earlier at 24 h. Subsequently, this expression decreased to basal levels but was expressed again in the 72 and 96 h post-inoculation samples. However, in the last sampling, *OsTOM1* transcriptional expression returned to basal conditions (Figure 6b). For its part, *OsYSL15* showed greater induction in its expression in plants inoculated with yeast, under Fe-sufficient conditions, at 72 and 120 h (Figure 6c) while, in the case of plants grown under Fe-deficient conditions, inoculation with *D. hansenii* produced an earlier activation of transcriptional expression at 24 h, which reached its highest peak at 72 and 96 h. Only a moment of decline in gene expression was observed at 48 h (Figure 6d). Finally, *OsIRT1* activation was more erratic under both conditions. Inoculation with *D. hansenii* induced its expression to significantly higher levels relative to the non-inoculated control at 24, 72, and 120 h under Fe-sufficient conditions (Figure 6e), and at 48, 72, 96, and 120 h under Fe-deficient conditions (Figure 6f).

### 2.3. Effect of Inoculation with the Bacterium P. simiae (Strain WCS417) or the Yeast D. hansenii (CBS 767) on Gene Expression Levels Related to Responses to Ethylene

Ethylene is a plant hormone directly involved in the regulation of many of the physiological and morphological responses of Strategy I plants, including rice plants, to Fe deficiency [32,34]. Hence, it is important to analyze the expression of some of the genes involved in the ethylene synthesis and transduction pathway in rice plants under the influence of the microorganisms analyzed in this study, such as *P. simiae* and *D. hansenii*.

#### 2.3.1. Effect of the Bacterium *P. simiae* (Strain WCS417) on the Expression of the Genes *OsEIN2*, *OsACS2*, *OsACO3*

Figure 7 shows the expression levels of genes related to the ethylene pathway in rice plants inoculated with the *P. simiae* strain WCS417. In the case of *OsEIN2*, involved in ethylene signaling, no induction was detected under Fe-sufficient conditions compared to control plants (Figure 7a). However, under Fe deficiency, clear activation was observed, maintaining expression levels significantly higher than controls throughout the monitoring period, with the sole exception of the 72 h sampling (Figure 7b). *n* regard to the genes encoding ethylene synthesis enzymes, *OsACS2* and *OsACO3*, differential behavior was evident. For *OsACS2*, expression fluctuated under both Fe sufficiency and deficiency conditions, but with occasional inductions compared to the control. In plants inoculated and grown with Fe, expression showed an early increase at 24 h, followed by a decrease and renewed transcriptional activation at 96 h (Figure 7c). In contrast, under Fe deficiency, an initial induction was also observed at 24 h, reaching a higher peak at 48 h (Figure 7d). For its part, *OsACO3* only showed a significant induction compared to the control under Fe deficiency conditions, highlighting higher expression values during the first 72 h of the experiment (Figure 7e,f).

#### 2.3.2. Effect of the Bacterium *D. hansenii* (Strain CBS767) on the Expression of the Genes *OsEIN2*, *OsACS2*, *OsACO3*

Regarding the role played by the yeast *D. hansenii* in the induction or lack thereof of the ethylene hormone, an inductive effect was also observed, as shown in Figure 8. Under Fe-sufficient conditions, *OsEIN2* was expressed very markedly and significantly higher than the control at 48 and 120 h (Figure 8a). However, under Fe-deficient conditions, *OsEIN2* showed similar expression levels at all sampling times, being significantly higher than the control only at 48 h (Figure 8b). In the case of *OsACS2*, the gene encoding the ACC synthase enzyme, an early inductive effect by the yeast *D. hansenii* was observed, followed by a decline, finally reaching the highest degree of induction of expression at 120 h after inoculation in plants grown under Fe-sufficient conditions (Figure 8c). Under Fe-deficient conditions, expression was found significantly higher than the control only at 48 h (Figure 8d). Regarding the *OsACO3* gene, responsible for encoding the ACC oxidase enzyme, under Fe-sufficient conditions, expression levels were only significantly higher than the control at 120 h (Figure 8e). However, under Fe-deficient conditions, it was significantly induced at 48 and 72 h (Figure 8f).

## 3. Discussion

The importance of phytosiderophores (PS) in modern agriculture lies not only in their role in plant nutrition, but also in their ecological implications by improving Fe solubility in alkaline and calcareous soils [3]. Although primarily associated with grasses, several studies have shown that certain microorganisms can influence their production. When interacting with plants, these microorganisms stimulate the host to increase PS synthesis or release their own siderophores with analogous functions in Fe chelation and mobilization [25]. Rice, despite being a grass, combines traits of Strategies I and II for Fe acquisition, showing versatility in the face of Fe deficiency [20,35]. A well-documented example of Fe^2+^ uptake in rice is the functionality of the *OsIRT1* transporter [36].

The results obtained (Figure 1 and Figure 2) show that both *Pseudomonas simiae* WCS417 and *Debaryomyces hansenii* CBS767 are capable of inducing PS production in rice plants. In the case of *P. simiae*, the inducing effect was observed under both Fe sufficiency and deficiency conditions, while in *D. hansenii* this effect was limited to Fe deficiency. These findings are consistent with previous research in which different *Pseudomonas* strains produce siderophores that modify Fe availability in the rhizosphere and stimulate PS production in the host [37]. Similarly, other bacteria such as *Bacillus* spp. have been described to promote rice growth by modulating mechanisms related to Fe acquisition [38]. Likewise, arbuscular mycorrhizal fungi have been shown to enhance Fe uptake by increasing PS production or through alternative solubilization mechanisms [39]. Moreover, there is evidence that showed that inoculation of rice with the non-pathogenic strain of the fungus *Fusarium oxysporum* FO12 increases PS exudation under Fe-deficiency conditions [40], highlighting the potential for the induction of these responses in cereals by different microorganisms. It is noteworthy that quantification of phytosiderophores was performed using the indirect method of Inal et al. [41], which estimates the Fe(III)-mobilizing capacity of root exudates. Although this technique does not discriminate between plant and microbe derived siderophores, it provides a robust and widely used comparative approach for evaluating Fe-deficiency responses in graminaceous plants under controlled hydroponic conditions. Both *P. simiae* and *D. hansenii* are known to produce siderophores when Fe is scarce, which may transiently modify Fe availability and contribute to activating Fe-acquisition pathways in the plant. Nevertheless, previous studies have shown that even siderophore-deficient *Pseudomonas* mutants can improve Fe uptake in plants [42], indicating that the observed effects mainly arise from the activation of plant Fe-deficiency signaling rather than direct microbial chelation.

At the molecular level, PS-related responses in monocots are regulated by the transcription factor *OsIRO2*, which controls the induction of genes involved in the biosynthesis and transport of these compounds [23]. In this study, both microorganisms significantly induced the expression of key genes such as *OsNAAT* and *OsIRO2*, as well as transporters *OsTOM1*, *OsYSL15*, and *OsIRT1* (Figure 4, Figure 5, Figure 6 and Figure 7). This confirms that microbial inoculation activates core components of the PS synthesis and transport machinery in rice. Similar results have been described in dicots, where inoculation with *Paenibacillus polymyxa* increased the expression of *FIT*, *FRO2*, and *IRT1* in *Arabidopsis* [43]. Similarly, *P. simiae* has been reported to induce the expression of Fe deficiency-related genes in *Arabidopsis* roots [44,45]. In the *D. hansenii* treatment, phytosiderophore production was induced only under Fe-deficient conditions. However, the qRT-PCR analysis revealed that several genes involved in phytosiderophore synthesis and Fe acquisition such as *OsNAAT*, *OsTOM1*, *OsYSL15*, and *OsIRT1* were upregulated under both Fe-sufficient and Fe-deficient conditions, although with noticeable differences in the timing of induction. From our perspective, a plausible explanation for the induction of phytosiderophore-related genes under Fe sufficiency, despite the observed reduction in phytosiderophore release, lies in the fact that gene expression does not necessarily correlate directly with the enzymatic activity of the proteins they encode. Indeed, post-transcriptional regulatory mechanisms, including those affecting protein stability, processing, or activation, are well documented [46]. Such regulatory layers may modulate enzyme functionality and thereby account for the lack of a strict correspondence between transcript accumulation and phytosiderophore production. This consideration may help explain why gene expressions related to phytosiderophore biosynthesis can be induced under Fe-sufficient conditions, whereas no detectable increase in phytosiderophore release is observed using analytical approaches such as that described by Inal et al. [41]. Future work in our group will focus on elucidating this discrepancy by examining protein abundance and activity to establish a more comprehensive understanding of the regulatory steps controlling phytosiderophore biosynthesis.

It is noteworthy that transcriptional expression showed oscillations over time, with activations and decreases at different sampling points. This pattern is characteristic of plant stress responses, where there are early-responding genes and others that activate later, regulated by feedback and metabolic competition (e.g., between PS and ethylene synthesis from SAM). This behavior has already been described in transcriptomic studies of iron deficiency and other stresses [40,43,47,48]. The role of *OsIRO2* extends beyond the regulation of Fe-acquisition genes, as this transcription factor belongs to the bHLH family that integrates Fe deficiency responses with hormonal signaling and ISR pathways [49]. Similar to the *FIT-bHLH38/bHLH39* module described in dicots, *OsIRO2* functions as a central node that coordinates the activation of genes such as *OsNAAT1*, *OsTOM1*, and *OsYSL15* in response to both Fe limitation and signaling molecules like ethylene or jasmonate. The upregulation of *OsIRO2* observed in the presence of *P. simiae* and *D. hansenii* therefore suggests that these microorganisms can trigger ISR-like signaling networks in rice roots, promoting Fe mobilization and improving plant resilience to nutrient stress.

Fe acquisition mechanisms are subject to hormonal regulation, with ethylene being one of the central signals. In Strategy I plants, ethylene regulates proton extrusion, ferric reductase activity, and root hair proliferation under Fe-deficient conditions [33,49]. The results of this work (Figure 8) suggest that this hormone could also mediate microbial effects in rice. *P. simiae* significantly induced *OsEIN2*, *OsACS2*, and *OsACO3* genes under Fe deficiency, whereas *D. hansenii* showed early or late activations depending on Fe availability. Considering that S-adenosylmethionine (SAM) is a precursor in both PS biosynthesis and ethylene production [3,50], it is plausible that microbial signals modulate the balance between both pathways, thus reinforcing responses to Fe deficiency. This integrative role highlights the possibility that ethylene acts as a central node connecting microbial signals with the activation of Strategy I and II mechanisms in rice.

The mechanisms underlying the induction of Fe-deficiency responses by microbial inoculation are likely multifactorial. In the case of *P. simiae* WCS417, previous studies have demonstrated the production of siderophores, volatile organic compounds (VOCs), and phytohormone-related metabolites capable of activating root Fe-acquisition genes [45,51,52]. These signals can transiently decrease Fe availability in the rhizosphere or stimulate hormonal pathways (ethylene and jasmonate) associated with Fe uptake. Although less studied in the rhizosphere, some yeasts are known to produce siderophores and VOCs [53,54], which may contribute to modifying Fe solubility and plant signaling or secrete cell wall-degrading enzymes, such as carboxymethyl cellulase, protease, or chitinase, to inhibit pathogenic growth [54]. In this context, *D. hansenii* has been proposed to be a microorganism that can act both as a growth promoter in plants of industrial interest [55,56] and as a biocontrol agent through the production of killer toxins active against a wide range of fungi [57,58,59]. However, while substantial information is available for bacterial models such as *P. simiae*, further studies are still required to elucidate the specific signaling mechanisms involved in the interaction between *D. hansenii* and plants before drawing definitive conclusions about its mode of action.

Taken together, the results obtained demonstrate that the rhizobacteria *P. simiae* and the halotolerant yeast *D. hansenii* can induce iron deficiency responses in rice through the production of PS and the activation of associated molecular pathways. While *P. simiae* exerts its effect under both Fe sufficiency and deficiency, *D. hansenii* does so primarily under deficiency conditions. The concomitant induction of ethylene-related genes reinforces the idea that microbial inoculation can harness hormonal signaling to enhance nutrient acquisition. Future studies should evaluate whether these microorganisms, applied individually or in combination, are capable of improving iron nutrition and rice yield under field conditions, and whether their effects are maintained in different soil types and under different environmental scenarios.

## 4. Materials and Methods

### 4.1. Plant Material and Growing Conditions

Rice plants (*Oryza sativa* L. var. ‘Puntal’) were used. The seeds were sterilized according to the protocol described by Aparicio et al. [33]. Subsequently, they were sown on a layer of moist perlite to which 20 mL of a 5 mM CaCl_2_ solution was added. The seeds were covered with another layer of perlite and kept in darkness at 27 °C for 4 days. After germination, the seedlings were transferred to a growth chamber with a 14 h photoperiod, 70% relative humidity, a temperature of 25 °C (day) and 22 °C (night), and irradiance of 300 μmol m^−2^ s^−1^, for 7 days. For hydroponic cultivation, seedlings were placed in foam supports inserted into polyurethane plates floating on the continuously aerated R&M nutrient solution [60]. The nutrient solution used was Römheld & Marschner [60] (2 mM Ca(NO_3_)_2_, 0.75 mM K_2_SO_4_, 0.65 mM MgSO_4_, 0.5 mM KH_2_PO_4_, 50 µM KCl, 10 µM H_3_BO_3_, 1 µM MnSO_4_, 0.5 µM CuSO_4_, 0.5 µM ZnSO_4_, 0.05 µM (NH_4_)_6_Mo_7_O_24_ and 45 µM Fe-EDTA). Each treatment consisted of five independent replicates, with eight plants per replicate. The nutrient solution was continuously aerated to prevent anoxia. The plants remained in these conditions until 22–25 days after sowing, at which time the treatments were applied.

For Fe-deficiency treatments, Fe-EDTA was omitted from the solution. Plants were grown for 22–25 days under these hydroponic conditions before the start of microbial inoculation.

### 4.2. Microbial Cultures and Plant Inoculation

The microorganisms (*D. hansenii* CBS767 and *P. simiae* WCS417) were maintained in stock in 20% glycerol at −80 °C.

For *P. simiae* WCS417, cells were plated on King’s B (KB) agar and incubated at 30 °C for 24 h. Colonies were recovered by suspension in 10 mM MgSO_4_ and centrifugation at 4500 rpm for 5 min. After two washes, the cell pellet was resuspended in the same solution, and the optical density was adjusted to OD_600_ = 0.8 before inoculation, corresponding to approximately 1 × 10^8^ CFU/mL, as verified by serial dilution and plating. The KB medium was prepared with peptone (2%), K_2_HPO_4_ (11.5 mM), agar (1.5%), and glycerol (1.4%), supplemented with MgSO_4_ (6.1 mM) and rifampicin (50 μg/mL) as a selective antibiotic.

The yeast *D. hansenii* (CBS767) was grown in YPD medium (2% glucose, 1% yeast extract, and 2% peptone) at 26 °C with shaking. The inoculum was prepared from these cultures by adjusting the cell suspension to 10^7^ cells/mL in sterile deionized water. and resuspended in the same solvent. Cell concentration was determined using a Neubauer counting chamber and adjusted to 10^7^ cells/mL before inoculation.

Inoculation with *P. simiae* and *D. hansenii* was performed by directly adding the suspensions (10^7^ cells/mL) to the nutrient solution, both in the presence (+Fe) and absence (−Fe) of iron. In hydroponic culture, the inoculum was incorporated into the nutrient solution to reach the desired concentration, whereas in perlite cultivation, plants were irrigated with the corresponding microbial suspension until the substrate reached field capacity, ensuring uniform distribution. The final microbial concentration in the growth medium was standardized to 10^7^ cells/mL in both systems to ensure comparable inoculation levels. Each treatment included its non-inoculated control. For subsequent gene expression analyses, plant roots were harvested and stored at −80 °C.

Inoculation was carried out on day 0, coinciding with the start of Fe-sufficient (+Fe) or Fe-deficient (−Fe) treatments. Sampling times (24–96 h for *P. simiae*; 24–120 h for *D. hansenii*) refer to hours post-inoculation.

### 4.3. Phytosiderophore Determinations

Phytosiderophore (PS) determinations were performed in plants grown in perlite, as this substrate allows the easy collection of root exudates. PS release into the rhizosphere was quantified following the methodology of Inal et al. [41] and Reichman and Parker [61] without modifications. In plants inoculated with *P. simiae*, determinations were made at 24, 48, 72, and 96 h after inoculation (time 0); in plants inoculated with *D. hansenii*, analyses were extended to 120 h. Fe sufficiency or deficiency treatments were applied simultaneously with inoculation. In the case of inoculation with *D. hansenii*, previous experiments showed a high level of gene expression at 96 h. In this study, we extended the observation period by one additional day to determine whether those expression levels were sustained or, as is commonly observed, began to decline [30].

### 4.4. Gene Expression Analysis by qRT-PCR

The relative expression of genes involved in phytosiderophore biosynthesis (*OsNAAT*, *OsIRO2*) and transport (*OsTOM1*, *OsYSL15*, *OsIRT1*), as well as genes related to the ethylene pathway (*OsEIN2*, *OsACS2*, *OsACO3*), were determined by qRT-PCR using the primers pairs shown on Table 1.

For gene expression analyses, plants were grown under hydroponic conditions using the same nutrient solution described in Section 4.1. At each sampling time (24, 48, 72, 96, and 120 h after inoculation), entire root systems were carefully collected, rinsed with cold distilled water to remove residual solution, frozen immediately in liquid nitrogen, and stored at −80 °C until RNA extraction. For each treatment and time point, three independent biological replicates were analyzed, each consisting of pooled roots from multiple plants. RNA extraction was carried out using Tri Reagent (Molecular Research Center, Inc., Cincinnati, OH, USA) following the manufacturer’s protocol. RNA concentration was measured at 260 nm. M-MLV reverse transcriptase (Promega, Madison, WI, USA) was used to generate cDNA from 3 μg of DNase-treated root RNA, using random hexamers for amplification. The study of gene expression by qRT-PCR was performed by using a qRT-PCR Bio-Rad CFX connect thermal cycler. The amplification profile consisted of cycles with the following conditions: initial denaturation and polymerase activation (95 °C for 3 min), amplification and quantification (90 °C for 10 s, 57 °C for 15 s, and 72 °C for 30 s), and a final melting curve stage of 65 to 95 °C with an increment of 0.5 °C for 5 s to ensure the absence of primer dimer or nonspecific amplification products. PCR reactions were set up in 20 μL of SYBR Green Bio-RAD PCR Master Mix, following the manufacturer’s instructions. Controls containing water instead of cDNA were included to detect contamination in the reaction components. Normalization was performed using a reference gene (*OsActin*). Normalization was performed against the reference housekeeping gene, and relative expression levels were calculated using the 2^−ΔΔC^ method, taking the expression level of the corresponding non-inoculated control under the same Fe condition as the calibrator.

### 4.5. Statistical Analysis

Phytosiderophore determinations were performed using five biological replicates per treatment (*n* = 5), while gene-expression analyses were conducted using three independent biological replicates, each analyzed in two technical qPCR replicates. Normality and homogeneity of variances were verified before applying statistical analyses. When necessary, data were transformed. Gene expression and PS production between inoculated treatments and controls were compared using one-way ANOVA and Dunnett’s test (*p* < 0.05). In some cases, pairwise comparisons were performed using Student’s t test or Mann–Whitney test (*p* < 0.05), as appropriate.

## 5. Conclusions

In this study, we demonstrate that both the rhizospheric bacterium *Pseudomonas simiae* WCS417 and the halotolerant yeast *Debaryomyces hansenii* CBS767 act as inducers of Fe deficiency responses in hydroponic rice plants. In *P. simiae*, phytosiderophore (PS) induction was observed under both Fe sufficiency and deficiency conditions, whereas in *D. hansenii*, this effect was mainly restricted to deficiency conditions.

Both microorganisms promoted the expression of genes related to PS biosynthesis (*OsNAAT* and *OsIRO2*) and transport (*OsTOM1*, *OsYSL15*, and *OsIRT1*), reinforcing the hypothesis that they are capable of activating central mechanisms of Fe acquisition in rice. Transcriptional expression showed oscillations over time, reflecting a dynamic typical of stress responses, where early and late response genes coexist, regulated by feedback processes and metabolic competition.

Finally, the induction of genes related to the ethylene pathway in inoculated plants suggests that this hormone acts as a key regulator of the response mechanisms to iron deficiency in rice, a species that combines elements of strategies I and II of iron acquisition. These results position *P. simiae* and *D. hansenii* as promising candidates for the development of biofertilizers aimed at improving the iron nutrition of crops in calcareous and alkaline soils.

## Figures and Tables

**Figure 1 plants-14-03769-f001:**
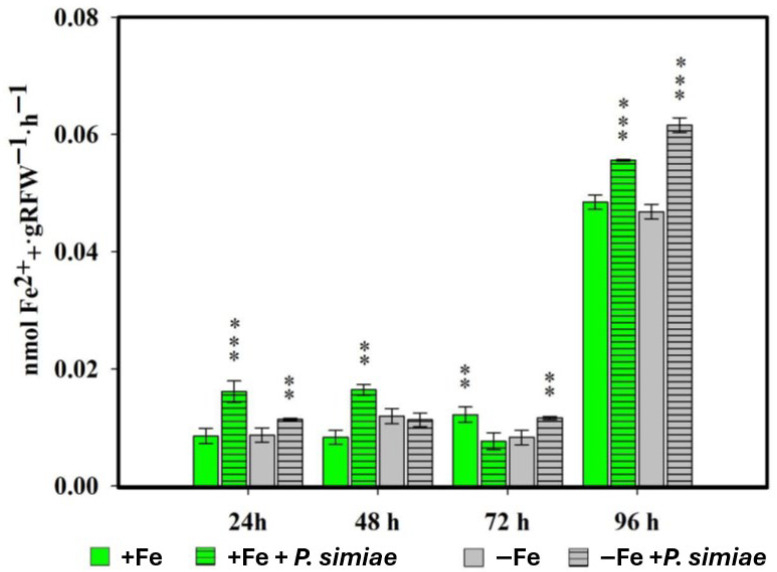
Phytosiderophore production in rice plants inoculated with *P. simiae* (WCS417) under iron sufficiency (+Fe) or deficiency (−Fe). Values are mean ± SE (*n* = 5). Asterisks indicate significant differences compared to the control (** *p* < 0.01, *** *p* < 0.001).

**Figure 2 plants-14-03769-f002:**
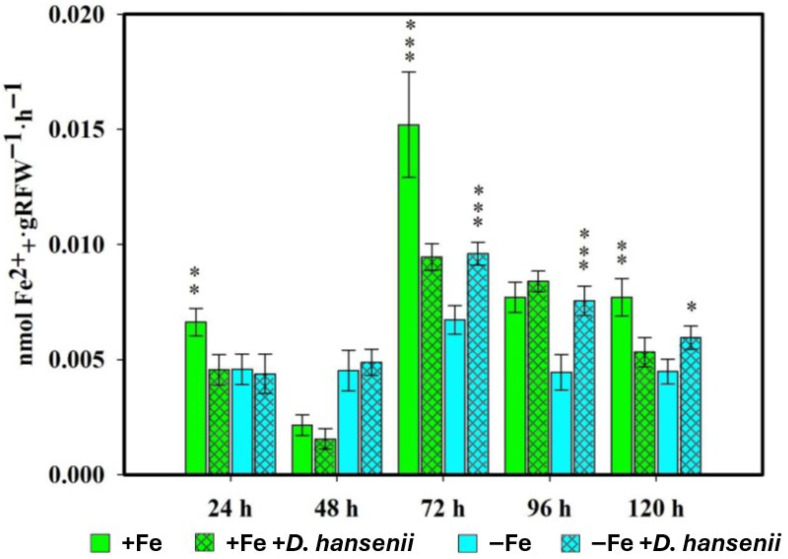
Phytosiderophore production in rice plants inoculated with *D. hansenii* (CBS767) under iron sufficiency (+Fe) or deficiency (−Fe). Values are mean ± SE (*n* = 5). Asterisks indicate significant differences compared to the control (* *p* < 0.05, ** *p* < 0.01, *** *p* < 0.001).

**Figure 3 plants-14-03769-f003:**
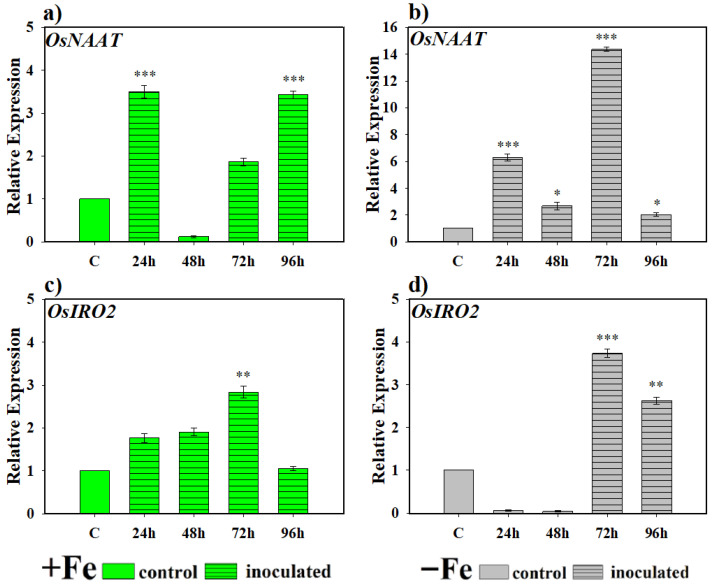
Expression of genes related to phytosiderophore (PS) production in rice plants inoculated with the bacterium *P. simiae* (WCS417). The experiments were carried out under hydroponic culture conditions in a growth chamber. Determinations were made at 24, 48, 72, and 96 h. Expression values are relative to the corresponding non-inoculated control under the same Fe condition. Treatments: +Fe = Fe sufficiency in the nutrient solution; +Fe + *P. simiae* = Fe sufficiency in the nutrient solution plus inoculation of the plants with the bacteria; −Fe = Fe deficiency in the nutrient solution; and −Fe + *P. simiae* = Fe deficiency in the nutrient solution plus inoculation of the plants with the bacteria. (**a**,**c**) expression of the *OsNAAT* and *OsIRO2* genes under Fe sufficiency conditions, respectively. (**b**,**d**) expression of the *OsNAAT* and *OsIRO2* genes under Fe deficiency conditions, respectively. The data represent the mean ± SE of three independent biological replicates and two technical replicates. Asterisks indicate significant differences compared to the control (* *p* < 0.05, ** *p* < 0.01, *** *p* < 0.001).

**Figure 4 plants-14-03769-f004:**
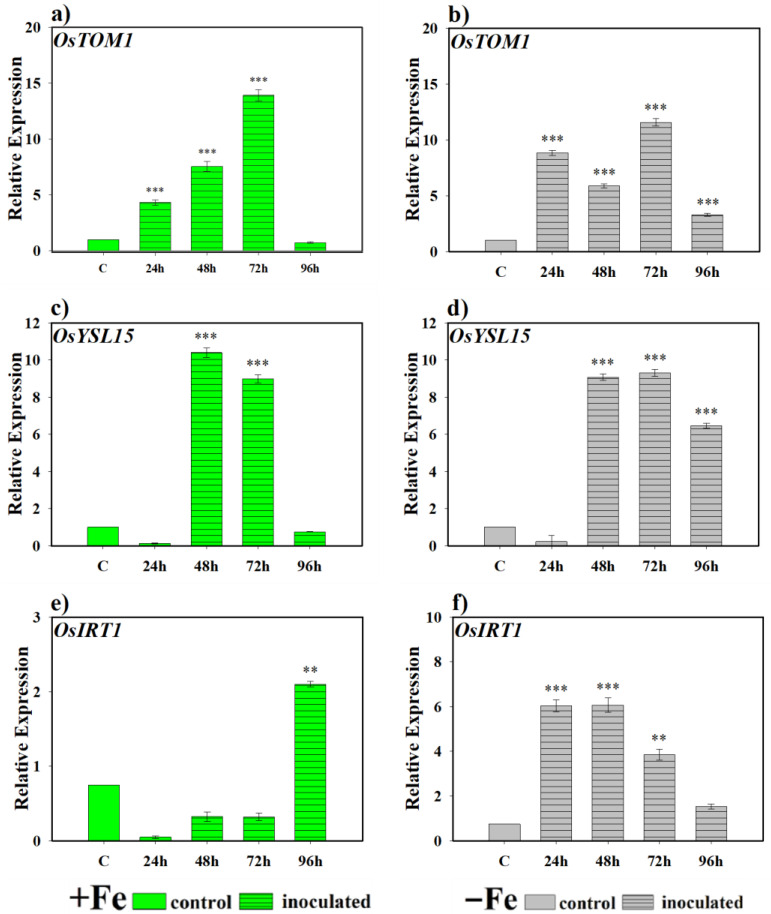
Expression of phytosiderophore (PS) transporter genes in rice plants inoculated with *P. simiae* (WCS417). The experiments were carried out under hydroponic culture conditions in a growth chamber. Determinations were made at 24, 48, 72, and 96 h. Expression values are relative to the corresponding non-inoculated control under the same Fe condition. Treatments: +Fe = Fe sufficiency in the nutrient solution; +Fe + *P. simiae* = Fe sufficiency in the nutrient solution plus inoculation of the plants with the bacteria; −Fe = Fe deficiency in the nutrient solution; and −Fe + *P. simiae* = Fe deficiency in the nutrient solution plus inoculation of the plants with the bacteria. (**a**,**c**,**e**) expression of the *OsTOM1*, *OsYSL15*, and *OsIRT1* genes under Fe sufficiency conditions, respectively. (**b**,**d**,**f**) expression of the *OsTOM1*, *OsYSL15* and *OsIRT1* genes under Fe deficiency conditions, respectively. The data represent the mean ± SE of three independent biological replicates and two technical replicates. Asterisks indicate significant differences compared to the control (** *p* < 0.01, *** *p* < 0.001).

**Figure 5 plants-14-03769-f005:**
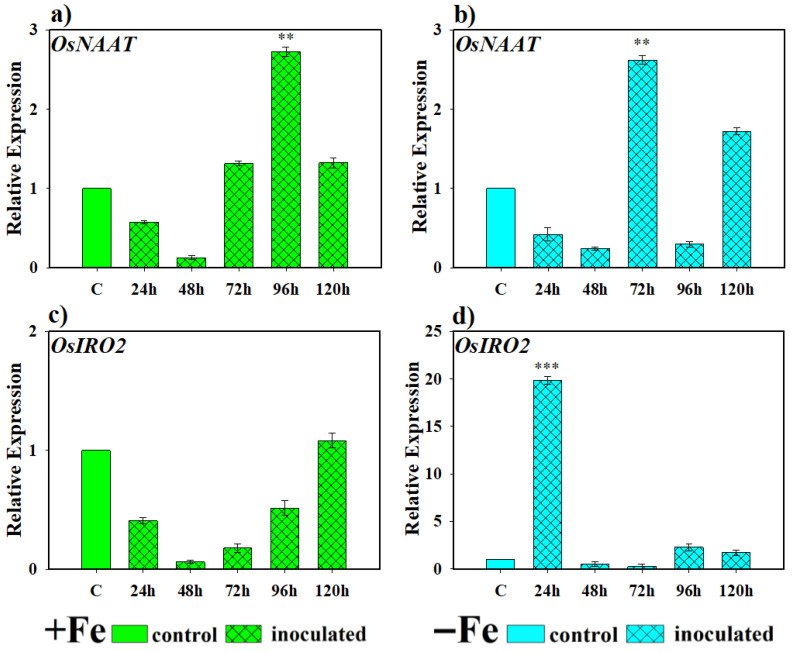
Expression of genes related to phytosiderophore (PS) production in rice plants inoculated with *D. hansenii* (CBS767). The experiments were carried out under hydroponic culture conditions in a growth chamber. Determinations were made at 24, 48, 72, and 96 h. Expression values are relative to the corresponding non-inoculated control under the same Fe condition. Treatments: +Fe = Fe sufficiency in the nutrient solution; +Fe + *P. simiae* = Fe sufficiency in the nutrient solution plus inoculation of the plants with the yeast; −Fe = Fe deficiency in the nutrient solution; and −Fe + *D. hansenii* = Fe deficiency in the nutrient solution plus inoculation of the plants with the yeast. (**a**,**c**) expression of the *OsNAAT* and *OsIRO2* genes under Fe sufficiency conditions, respectively. (**b**,**d**) expression of the *OsNAAT* and *OsIRO2* genes under Fe deficiency conditions, respectively. The data represent the mean ± SE of three independent biological replicates and two technical replicates. Asterisks indicate significant differences compared to the control (** *p* < 0.01, *** *p* < 0.001).

**Figure 6 plants-14-03769-f006:**
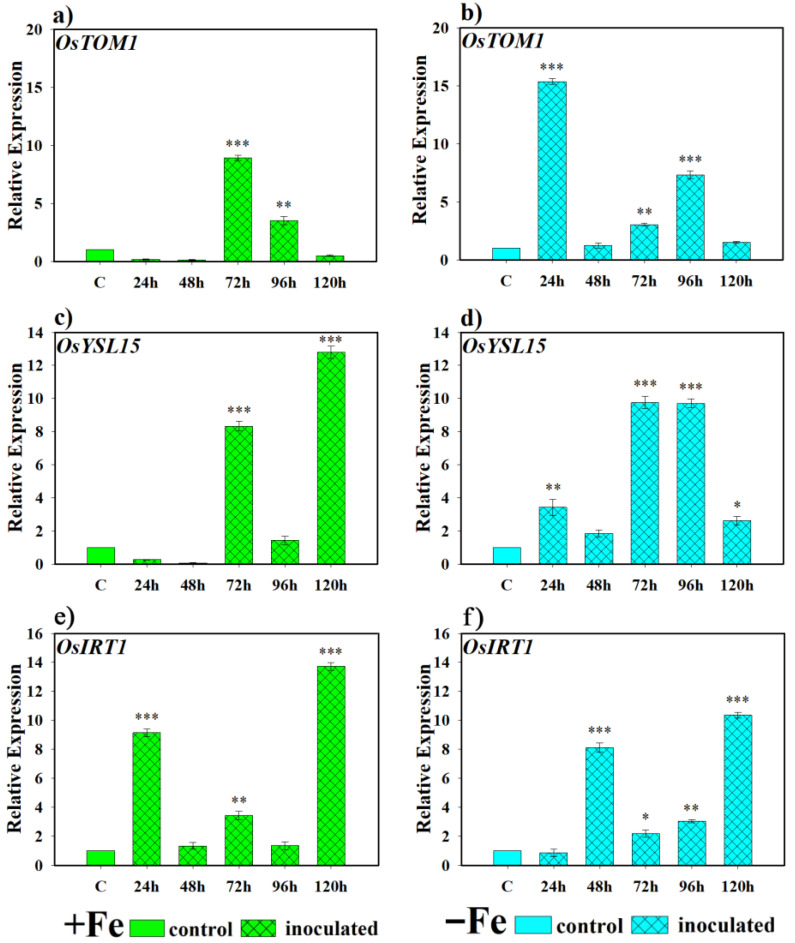
Expression of phytosiderophore (PS) transporter genes in rice plants inoculated with *D. hansenii* (CBS767). The experiments were carried out under hydroponic culture conditions in a growth chamber. Determinations were made at 24, 48, 72, and 96 h. Expression values are relative to the corresponding non-inoculated control under the same Fe condition. Treatments: +Fe = Fe sufficiency in the nutrient solution; +Fe + *D. hansenii* = Fe sufficiency in the nutrient solution plus inoculation of the plants with the yeast; −Fe = Fe deficiency in the nutrient solution; and −Fe + *D. hansenii* = Fe deficiency in the nutrient solution plus inoculation of the plants with the yeast. (**a**,**c**,**e**) expression of the *OsTOM1*, *OsYSL15*, and *OsIRT1* genes under Fe sufficiency conditions, respectively. (**b**,**d**,**f**) expression of the *OsTOM1*, *OsYSL15* and *OsIRT1* genes under Fe deficiency conditions, respectively. The data represent the mean ± SE of three independent biological replicates and two technical replicates. Asterisks indicate significant differences compared to the control (* *p* < 0.05, ** *p* < 0.01, *** *p* < 0.001).

**Figure 7 plants-14-03769-f007:**
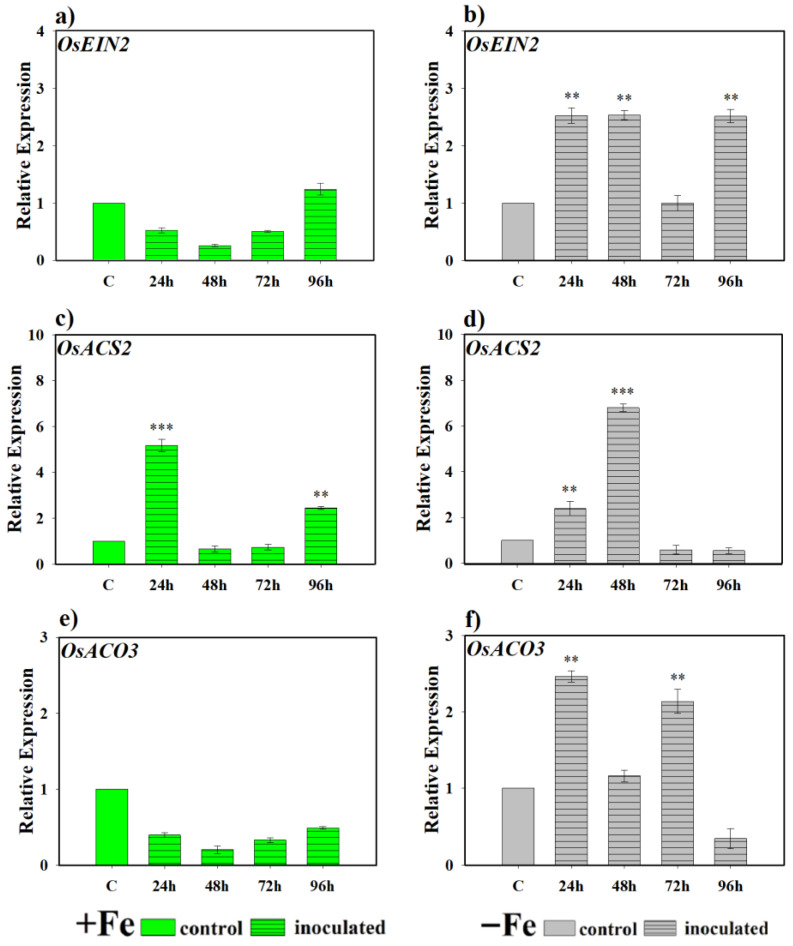
Effect of *P. simiae* (WCS417) on the expression of ethylene-related genes in rice roots. Experiments were conducted under hydroponic culture conditions in a growth chamber. Determinations were performed at 24, 48, 72, and 96 h. Expression values are relative to the corresponding non-inoculated control under the same Fe condition. Treatments: +Fe = iron-sufficient nutrient solution; +Fe + *P. simiae* = iron-sufficient nutrient solution plus bacterial inoculation; −Fe = iron-deficient nutrient solution; and −Fe + *P. simiae* = iron-deficient nutrient solution plus bacterial inoculation. (**a**,**b**) Expression of the *OsEIN2* gene involved in ethylene signal transduction under Fe sufficiency and deficiency, respectively. (**c**,**e**) Expression of *OsACS2* and *OsACO3* genes involved in ethylene biosynthesis under Fe sufficiency, respectively. (**d**,**f**) Expression of *OsACS2* and *OsACO3* genes involved in ethylene biosynthesis under Fe deficiency, respectively. The data represent the mean ± SE of three independent biological replicates and two technical replicates. Asterisks indicate significant differences compared to the control (** *p* < 0.01, *** *p* < 0.001).

**Figure 8 plants-14-03769-f008:**
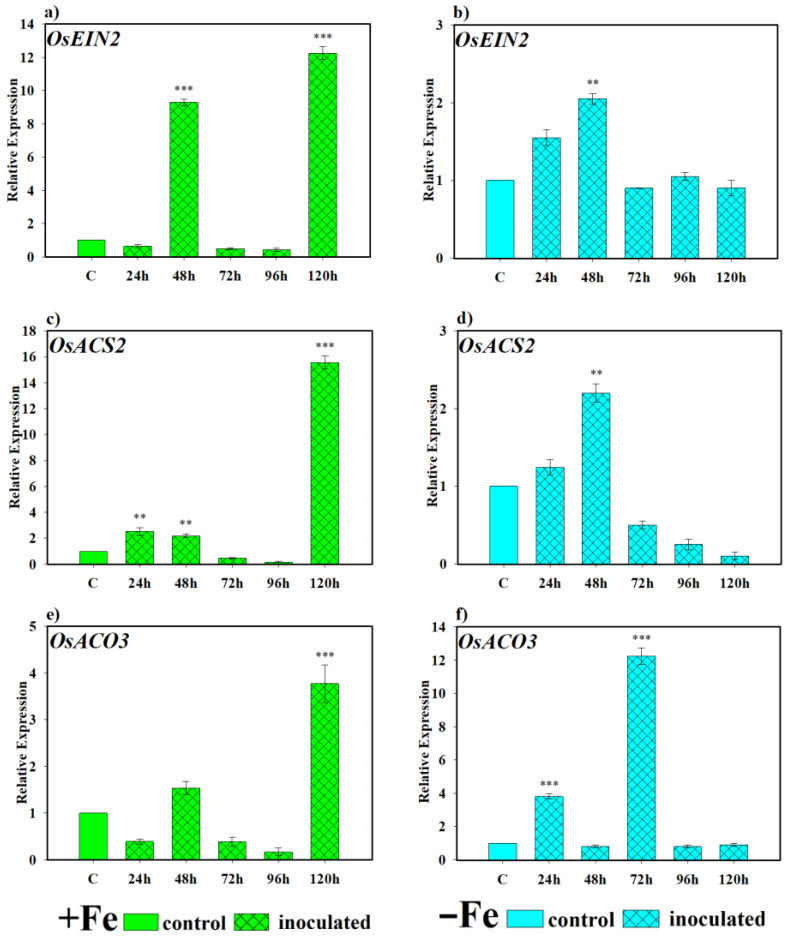
Effect of *D. hansenii* (CBS767) on the expression of ethylene-related genes in rice roots. Experiments were conducted under hydroponic culture conditions in a growth chamber. Determinations were performed at 24, 48, 72, 96, and 120 h. Expression values are relative to the corresponding non-inoculated control under the same Fe condition. Treatments: +Fe = iron-sufficient nutrient solution; +Fe + *D. hansenii* = iron-sufficient nutrient solution plus yeast inoculation; −Fe = iron-deficient nutrient solution; and −Fe + *D. hansenii* = iron-deficient nutrient solution plus yeast inoculation. (**a**,**b**) Expression of the *OsEIN2* gene involved in ethylene signal transduction under Fe sufficiency and deficiency, respectively. (**c**,**e**) Expression of *OsACS2* and *OsACO3* genes involved in ethylene biosynthesis under Fe sufficiency, respectively. (**d**,**f**) Expression of *OsACS2* and *OsACO3* genes involved in ethylene biosynthesis under Fe deficiency, respectively. The data represent the mean ± SE of three independent biological replicates and two technical replicates. Asterisks indicate significant differences compared to the control (** *p* < 0.01, *** *p* < 0.001).

**Table 1 plants-14-03769-t001:** Primer pairs used for analysis of rice gene expression by qRTPCR.

Gene	Forward (5′–3′)	Reverse (5′–3′)
*OsNAAT1*	TAAGAGGATAATTGATTTGCTTAC	CTGATCATTCCAATCCTAGTACAAT
*OsIRO2*	CTCCCATCGTTTCGGCTACCT	GCTGGGCACTCCTCGTTGATC
*OsTOM1*	GCCCAAGAACGCCAAAATGA	GGCTTGAAGGTCAACGCAAG
*OsYSL15*	AACATAAGGGGGACTG GTAC	TGATTACCGCAATGATGCTTAG
*OsIRT1*	CGTC TTCTTCTTCTCCACCACGAC	GCAGCTGATGATCGAGTCTGACC
*OsEIN2*	GCTGCGGTAGAGAAGCTATT	TGTACTGGATGTCTGCCTTATC
*OsACS2*	TTTGGCGCCTTGACGGCCTC	AAAGGGAGCGCACCATGGCC
*OsACO3*	TGCAACAGCACGCCACACCA	TGGATCGACGTCCAGCCCGT
*OsActin*	TGCTATGTACGTCGC CATCCAG	AATGAGTAACCACGCTCCGTCA

## Data Availability

The original contributions presented in the study are included in the article. All the data included in this article are publicly available. Further inquiries can be directed to the corresponding author.

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
