# Peer review of "Pseudomonas simiae WCS417 and Debaryomyces hansenii Induce Iron Deficiency Responses in Rice (Oryza sativa L.) Through Phytosiderophore Production and Gene Expression Modulation"

_plants, 2025, doi:10.3390/plants14243769_

Round 1
Reviewer 1 Report
Comments and Suggestions for Authors
This article continues the research cycle of the author team on studying the mechanisms of plant response to iron deficiency. Recently, the positive effect of inoculation with a non-pathogenic fungus Fusarium oxysporum FO12 on the production of phytosiderophores by rice plants was established (Plants 2023, 12, 3145). The authors evaluated two other microbial cultures, bacteria Pseudomonas simiae WCS417 and yeast Debaryomyces hansenii CBS767, which were known to be PGPB but were not considered in relation to their involvement in providing plants with iron. Based on experimental results, it was concluded that these microorganisms may be promising for use in agribiotechnology to increase iron availability for rice plants.
When reviewing the article, I had a number of questions regarding both the methodology and the discussion of the results obtained. I believe that in its current form, the article cannot be recommended for publication.
Major comments:
- The experimental methods are described extremely inadequately. This prevents an assessment of the results obtained and their reliability.
In section 4.1, there is no data on the composition of the nutrient substrate for either perlite cultivation or hydroponics, including the iron content and form. How many plants were used? The plant growth was performed at different times, and the growing periods differ - 7 days in perlite and 22-25 days in hydroponics. When inoculation was done in the both experiments? What experiments were used for siderophores and gene expression evaluation? What does mean time periods of 24-96 (120) h? After inoculation?
It's unclear how inoculation was carried out in both the perlite and hydroponic systems. Only data on the number of yeast cells in the suspension used for inoculation are provided. How was it evaluated? What was cell concentration in the growth medium, was it the same in perlite and hydroponic experiments? For bacteria, even the optical density of the inoculum is not specified (L.409).
There is no information on the number of replicates for all experiments and determinations, while there is a section 4.5 on the statistical analysis of the reliability of the results.
- When discussing the results, I would like to clarify the following points. If the reviewer understood correctly from the rambling description of the methods, the PS determination was conducted in an experiment with perlite, and gene expression was measured in hydroponics. This means that the plants differed in both their growing times and conditions. It needs to be justified why these definitions could not be established during a single experiment.
Differences in PS formation and genes expression were established under iron-present and iron-deficient conditions. What could be the significance of these processes for plants where there is a sufficient amount of iron in the environment?
Other points:
- Sections 4.2 and 4.3 have the same titles.
- Please specify which method was used for the quantitative assessment of phytosiderophores. Described in Inal et al 2007 paper or modified?
- 341 – Figures 1 and 2.
Author Response
Reviewer 1
Comments and Suggestions for Authors
This article continues the research cycle of the author team on studying the mechanisms of plant response to iron deficiency. Recently, the positive effect of inoculation with a non-pathogenic fungus Fusarium oxysporum FO12 on the production of phytosiderophores by rice plants was established (Plants 2023, 12, 3145). The authors evaluated two other microbial cultures, bacteria Pseudomonas simiae WCS417 and yeast Debaryomyces hansenii CBS767, which were known to be PGPB but were not considered in relation to their involvement in providing plants with iron. Based on experimental results, it was concluded that these microorganisms may be promising for use in agribiotechnology to increase iron availability for rice plants.
When reviewing the article, I had a number of questions regarding both the methodology and the discussion of the results obtained. I believe that in its current form, the article cannot be recommended for publication.
Major comments:
- The experimental methods are described extremely inadequately. This prevents an assessment of the results obtained and their reliability.
In section 4.1, there is no data on the composition of the nutrient substrate for either perlite cultivation or hydroponics, including the iron content and form. How many plants were used? The plant growth was performed at different times, and the growing periods differ - 7 days in perlite and 22-25 days in hydroponics. When inoculation was done in the both experiments? What experiments were used for siderophores and gene expression evaluation? What does mean time periods of 24-96 (120) h? After inoculation?
Author´s response: Thank for your comments. We have expanded each Section in material and methods to include a complete description of the methodology (nutrient solution composition, the form and concentration of Fe (Fe-EDTA 45 µM), number of plants per treatment, the timing of inoculation and sampling…). We also clarified that phytosiderophore production was determined in perlite-grown plants, while gene-expression analyses were performed in hydroponic culture. Sampling times (24–120 h) are now specified as hours after inoculation.
It's unclear how inoculation was carried out in both the perlite and hydroponic systems. Only data on the number of yeast cells in the suspension used for inoculation are provided. How was it evaluated? What was cell concentration in the growth medium, was it the same in perlite and hydroponic experiments? For bacteria, even the optical density of the inoculum is not specified (L.409).
Author´s response: We thank the reviewer for this helpful observation. The description of the inoculation procedure has been expanded in Section 4.2 to include detailed information on the preparation and standardization of the inoculum for both microorganisms and experimental systems.
There is no information on the number of replicates for all experiments and determinations, while there is a section 4.5 on the statistical analysis of the reliability of the results.
Author´s response: We appreciate this observation. Although the number of replicates was already indicated in each figure legend, we have now clarified this information in Section 4.7 (“Statistical analysis”) for completeness. The revised text specifies that phytosiderophore determinations were performed with five biological replicates (n = 5), and gene-expression analyses with three biological replicates and two technical qPCR replicates.
- When discussingthe results, I would like to clarify the following points. If the reviewer understood correctly from the rambling description of the methods, the PS determination was conducted in an experiment with perlite, and gene expression was measured in hydroponics. This means that the plants differed in both their growing times and conditions. It needs to be justified why these definitions could not be established during a single experiment. Differences in PS formation and genes expression were established under iron-present and iron-deficient conditions. What could be the significance of these processes for plants where there is a sufficient amount of iron in the environment?
Author´s response: We appreciate this valuable observation. As now clarified in the revised manuscript (Sections 4.3 and Discussion), phytosiderophore quantification was conducted in perlite-grown plants because this inert substrate allows the sterile and quantitative recovery of root exudates, whereas gene-expression analyses required hydroponic culture to ensure homogeneous Fe exposure and immediate root preservation for RNA extraction. Performing both analyses within a single experimental system was not technically feasible, since the requirements for exudate collection and RNA preservation are incompatible. Additionally, the observed induction of Fe-deficiency responses under Fe sufficiency has been discussed as a priming effect, where microbial inoculation activates early signaling pathways to enhance plant readiness for future Fe limitation (Zamioudis et al., 2015; Romera et al., 2019).
Other points:
- Sections 4.2 and 4.3 have the same titles.
Author´s response: Thank you for noticing this oversight. The section titles have been corrected to clearly differentiate both microorganisms:
- Please specify which method was used for the quantitative assessment of phytosiderophores. Described in Inal et al 2007 paper or modified?
Author´s response: We thank the reviewer for pointing out this clarification. The quantification of phytosiderophores was performed following exactly the protocol described by Inal et al. (2007), without any modification. Briefly, the method is based on the Fe(III)-mobilizing capacity of root exudates using Fe(OH)3 as the Fe source, and the released Fe(II) was quantified colorimetrically with bathophenanthroline disulfonate at 535 nm. The procedure was applied identically under both Fe-sufficient and Fe-deficient conditions to ensure full comparability among treatments.
- 341 – Figures 1 and 2.
Author´s response: Thank you for noticing this oversight. We have verified and corrected the figure numbering.
Reviewer 2 Report
Comments and Suggestions for Authors
Overall Summary:
This study reports that the rhizobacterium Pseudomonas simiae WCS417 and the halotolerant yeast Debaryomyces hansenii CBS767 can activate iron (Fe) deficiency responses in rice by inducing phytosiderophore (PS) production and modulating the expression of associated genes. The findings present a potentially novel insight and suggest a promising biotechnological approach for improving crop Fe nutrition using microbes. However, the manuscript, in its current form, has significant weaknesses in the depth of experimental design, the completeness of the data, and the strength of the conclusions. These shortcomings substantially limit the interpretation and impact of the study. Major revisions are necessary before it can be considered for publication.
- The study relies solely on PS quantification and qRT-PCR data, which represent only one level of biological response. A major limitation is the lack of corresponding physiological and phenotypic data from the plants. Without these, the biological relevance of the observed molecular changes remains unverified and speculative. It is essential to include: plant growth metrics:1) biomass (shoot and root dry/fresh weight), plant height, and tiller number for both inoculated and non-inoculated plants under Fe-sufficient and Fe-deficient conditions. These measurements provide direct evidence of any beneficial effect—if gene induction does not improve growth, its significance is greatly reduced. 2) Fe status: direct measurement of Fe content in shoots and roots is critical to confirm whether the reported induction of Fe acquisition mechanisms actually enhances Fe uptake. This data is necessary to support the manuscript’s central conclusion. 3) chlorophyll content: since Fe deficiency typically causes leaf chlorosis, SPAD values and/or leaf photographs should be provided to visually demonstrate whether microbial inoculation alleviates Fe deficiency symptoms.
- The study observes change in gene expression but does not identify the initial microbial signals responsible for these responses. It remains unclear whether siderophores, volatile organic compounds (VOCs), or other metabolites trigger the induction, or if the effect is direct or indirect. This fundamental mechanistic question is left unanswered, and more data are required to address this gap.
Minor Points:
There is an error in the figure legends for Figures 7 and 8, where OsACS2 is incorrectly written as OsACO2. This must be corrected.
Author Response
Reviewer 2
Comments and Suggestions for Authors
Overall Summary:
This study reports that the rhizobacterium Pseudomonas simiae WCS417 and the halotolerant yeast Debaryomyces hansenii CBS767 can activate iron (Fe) deficiency responses in rice by inducing phytosiderophore (PS) production and modulating the expression of associated genes. The findings present a potentially novel insight and suggest a promising biotechnological approach for improving crop Fe nutrition using microbes. However, the manuscript, in its current form, has significant weaknesses in the depth of experimental design, the completeness of the data, and the strength of the conclusions. These shortcomings substantially limit the interpretation and impact of the study. Major revisions are necessary before it can be considered for publication.
- The study relies solely on PS quantification and qRT-PCR data, which represent only one level of biological response. A major limitation is the lack of corresponding physiological and phenotypic data from the plants. Without these, the biological relevance of the observed molecular changes remains unverified and speculative. It is essential to include: plant growth metrics:1) biomass (shoot and root dry/fresh weight), plant height, and tiller number for both inoculated and non-inoculated plants under Fe-sufficient and Fe-deficient conditions. These measurements provide direct evidence of any beneficial effect—if gene induction does not improve growth, its significance is greatly reduced. 2) Fe status: direct measurement of Fe content in shoots and roots is critical to confirm whether the reported induction of Fe acquisition mechanisms actually enhances Fe uptake. This data is necessary to support the manuscript’s central conclusion. 3) chlorophyll content: since Fe deficiency typically causes leaf chlorosis, SPAD values and/or leaf photographs should be provided to visually demonstrate whether microbial inoculation alleviates Fe deficiency symptoms.
Author´s response: We acknowledge the reviewer’s valid concern regarding the inclusion of complementary physiological data. The aim of the present work was to characterize early molecular and biochemical responses induced by microbial inoculation, focusing on short-term mechanisms (phytosiderophore production and gene expression) that occur within the first 96–120 h after inoculation. At this stage, differences in biomass, Fe content, or chlorophyll levels are not yet pronounced. Although only early molecular responses were analyzed, these results are highly informative because the activation of genes involved in Fe acquisition precedes detectable changes in plant phenotype. Several studies have demonstrated that microbial-induced transcriptional activation of Fe-deficiency responses (including OsNAAT, OsIRO2, and OsIRT1) is an early indicator of improved Fe uptake and growth under prolonged Fe limitation (Zamioudis et al., 2015; Romera et al., 2019; Aparicio et al., 2023, Nuñez-Cano et al., 2025).
- The study observes change in gene expression but does not identify the initial microbial signals responsible for these responses. It remains unclear whether siderophores, volatile organic compounds (VOCs), or other metabolites trigger the induction, or if the effect is direct or indirect. This fundamental mechanistic question is left unanswered, and more data are required to address this gap.
Author´s Response: We appreciate the reviewer’s insightful comment. To address this point, we have expanded the Discussion section to include a new paragraph describing the potential mechanisms by which both microorganisms could induce Fe-deficiency responses. This addition summarizes current knowledge on the production of siderophores, volatile organic compounds (VOCs), phytohormone-related metabolites, and cell wall–degrading enzymes that may influence Fe solubility, plant signaling, and microbial interactions. Furthermore, the role of D. hansenii as a potential plant growth promoter and biocontrol agent has been highlighted, emphasizing its capacity to produce killer toxins active against a wide range of fungi. This modification strengthens the discussion by providing a broader mechanistic context for the observed plant responses and clarifying the potential roles of both microorganisms
Minor Points:
There is an error in the figure legends for Figures 7 and 8, where OsACS2 is incorrectly written as OsACO2. This must be corrected.
Author´s response: Thank you for your feedback. We've corrected the error.
Reviewer 3 Report
Comments and Suggestions for Authors
Comments to Manuscript” Pseudomonas simiae WCS417 and Debaryomyces hansenii CBS767 induce iron deficiency responses in rice (Oryza sativa L.) through phytosiderophore production and gene expression modulation“, submitted by Nuñez-Cano et al.
The manuscript deals with an interesting question, if rice plants can be supported by the two microorganisms Pseudomonas simiae WCS417 and Debaryomyces hansenii CBS767 to cope with iron deficiency conditions. The authors measured iron contents in the plants grown under –Fe and +Fe conditions with and without incoculation and found time-dependent differences in the contents, which suddenly increased dramatically after 92h when the bacterium was present, and after 72h when the yeast was pesent, for both iron conditions. They conclude from the expression profiles of a gene involved in phytosiderophore biosynthesis, of a iron-related bHLH transcription factor 2 gene, the genes of three transporters related to iron, and genes related to ethylene production, that the microorganisms stimulate phytosiderophore production. The phytosiderophore concentration was indirectly determined.
There are several major concerns, concerning the methods, and the interpretation/discussion of the results.
First it had to be mentioned that IRO2 is an iron-related bHLH transcription factor 2, as written in the discussion, (see also https://doi.org/10.1111/nph.16232) and not an enzyme acting in DMA biosynthesis as written in the results. It is necessary to give some short explanation regarding the exact functions of the proteins encoded by the genes related to DMA used for the expression studies. The up- and downs of transcripts are not well explained. In all figures dealing with expressions it is not clear what the authors mean with “control”. There is only one control (for what?), time? Is this an average value?
The methods are mostly adequate except for the 2′-deoxymugineic acid determination. The authors used a method described by Inal et al 2007: “Phytosiderophore (PS) was indirectly determined by calculating the amount of Fe(III) mobilized from Fe hydroxide with a modified method of Takagi”, which has limitations. I wonder why not a more accurate method for 2′-deoxymugineic acid was used, for instance the method described by Schindlegger et al., 2014. Siderophores are also produced by the bacterium, for instance pyoverdine, and pyochelin, and perhaps by the yeast. Moreover, it is known that Gram-negative bacteria degrade DMA within some hours. Generally, many siderophores are degraded by secreted bacterial enzymes. P. simiae is also catalase- and oxidase-positive. It is therefore important to demonstrate that the plant-secreted phytosiderophore increased in the medium. I doubt that an exact differentiation between microbial siderophores and the phytosiderophore is impossible with the method used by the authors. Moreover, iron contents in the plant can be influenced by phenolic compounds, ascorbate, oxalate and other small molecules, which can contribute to iron acquisition. This possibility is neither considered in the experiments nor discussed.
The fact, that the microorganisms might suffer from iron deficiency as well and could start to produce their own chelators and siderphores, was not taken into consideration. Under iron deficiency condition, microorganisms may develop an antagonistic behavior. When the requirements of the microorganisms are overseen, results can be easily misinterpreted. Does P. simiae start to produce its own siderophores? Unfortunately, the point was not addressed.
To my opinion, the study is not ready to get published.
Taken together, to my opinion the study is incomplete and needs at least correct DMA determinations.
Comments on the Quality of English LanguageThere are many type errors
Author Response
Reviewer 3
Comments and Suggestions for Authors
Comments to Manuscript” Pseudomonas simiae WCS417 and Debaryomyces hansenii CBS767 induce iron deficiency responses in rice (Oryza sativa L.) through phytosiderophore production and gene expression modulation“, submitted by Nuñez-Cano et al.
The manuscript deals with an interesting question, if rice plants can be supported by the two microorganisms Pseudomonas simiae WCS417 and Debaryomyces hansenii CBS767 to cope with iron deficiency conditions. The authors measured iron contents in the plants grown under –Fe and +Fe conditions with and without incoculation and found time-dependent differences in the contents, which suddenly increased dramatically after 92h when the bacterium was present, and after 72h when the yeast was pesent, for both iron conditions. They conclude from the expression profiles of a gene involved in phytosiderophore biosynthesis, of a iron-related bHLH transcription factor 2 gene, the genes of three transporters related to iron, and genes related to ethylene production, that the microorganisms stimulate phytosiderophore production. The phytosiderophore concentration was indirectly determined.
There are several major concerns, concerning the methods, and the interpretation/discussion of the results.
First it had to be mentioned that IRO2 is an iron-related bHLH transcription factor 2, as written in the discussion, (see also https://doi.org/10.1111/nph.16232) and not an enzyme acting in DMA biosynthesis as written in the results. It is necessary to give some short explanation regarding the exact functions of the proteins encoded by the genes related to DMA used for the expression studies. The up- and downs of transcripts are not well explained. In all figures dealing with expressions it is not clear what the authors mean with “control”. There is only one control (for what?), time? Is this an average value?
Author´s response: We thank the reviewer for this accurate observation. We agree that OsIRO2 is not an enzyme but an iron-related basic helix-loop-helix (bHLH) transcription factor, which positively regulates genes involved in the biosynthesis and transport of deoxymugineic acid (DMA) in rice, such as OsNAAT1, OsTOM1 and OsYSL15 (Ogo et al., 2007; Ogo et al., 2011). Accordingly, the text in the Results section has been corrected to reflect this regulatory role. We have also expanded the explanation of the molecular functions of the genes analyzed: OsNAAT1 encodes nicotianamine aminotransferase involved in DMA synthesis, OsTOM1 encodes the transporter responsible for DMA efflux, OsYSL15 encodes the transporter responsible for Fe(III)–DMA uptake, and OsIRT1 encodes an Fe(II) transporter expressed under iron deficiency.
Additionally, a short description of the OsIRO2 function and its relation with the ethylene-regulated network of Fe deficiency responses and the Induced Systemic Resistance (ISR) pathway was included (Kobayashi & Nishizawa, 2012; Romera et al., 2019).
The definition of “control” has also been clarified in the Methods and figure legends:
“Control” refers to the expression level of the same treatment (non-inoculated control under the same Fe condition), which was used as the calibrator for relative quantification (2-ΔΔCt method). Expression values at later time points are therefore represented relative to their own 0 h reference.
The methods are mostly adequate except for the 2′-deoxymugineic acid determination. The authors used a method described by Inal et al 2007: “Phytosiderophore (PS) was indirectly determined by calculating the amount of Fe(III) mobilized from Fe hydroxide with a modified method of Takagi”, which has limitations. I wonder why not a more accurate method for 2′-deoxymugineic acid was used, for instance the method described by Schindlegger et al., 2014. Siderophores are also produced by the bacterium, for instance pyoverdine, and pyochelin, and perhaps by the yeast. Moreover, it is known that Gram-negative bacteria degrade DMA within some hours. Generally, many siderophores are degraded by secreted bacterial enzymes. P. simiae is also catalase- and oxidase-positive. It is therefore important to demonstrate that the plant-secreted phytosiderophore increased in the medium. I doubt that an exact differentiation between microbial siderophores and the phytosiderophore is impossible with the method used by the authors. Moreover, iron contents in the plant can be influenced by phenolic compounds, ascorbate, oxalate and other small molecules, which can contribute to iron acquisition. This possibility is neither considered in the experiments nor discussed.
The fact, that the microorganisms might suffer from iron deficiency as well and could start to produce their own chelators and siderphores, was not taken into consideration. Under iron deficiency condition, microorganisms may develop an antagonistic behavior. When the requirements of the microorganisms are overseen, results can be easily misinterpreted. Does P. simiae start to produce its own siderophores? Unfortunately, the point was not addressed.
Author´s response: We appreciate the reviewer’s detailed and constructive comments. We agree that the method of Inal et al. (2007) provides an indirect estimation of phytosiderophores (PS), based on the Fe(III) mobilization capacity from Fe hydroxide, and that more sophisticated analytical approaches such as HPLC-MS detection of 2′-deoxymugineic acid (DMA) (e.g., Schindlegger et al., 2014) can offer higher specificity. However, our objective was not to quantify absolute DMA concentrations, but rather to obtain a comparative and complementary indication of PS production to support the expression results of OsNAAT1, OsIRO2, and OsTOM1.
The method used is widely adopted for functional studies of Fe-deficiency responses (e.g., Reichman & Parker, 2007; Inal et al., 2007; Kobayashi & Nishizawa, 2012) and offers reproducible results consistent with molecular data. As stated in the revised manuscript, PS determination was therefore considered a secondary but supporting parameter, while the main focus was the induction of Fe-deficiency-responsive genes by the microorganisms.
Regarding the possibility that the microorganisms themselves produce siderophores, we agree that this could partially contribute to the Fe mobilization measured. Both P. simiae and D. hansenii are known to secrete siderophore-like compounds under Fe-limiting conditions. However, previous studies have shown that even Pseudomonas mutants deficient in siderophore synthesis can still enhance Fe acquisition and chlorophyll levels in plants (Meziane et al., 2005). This indicates that microbial inoculation promotes Fe uptake not only through direct siderophore production, but also by activating plant signaling and Fe-acquisition pathways.
Indeed, our previous work demonstrated that both P. simiae and D. hansenii facilitate Fe assimilation in cucumber and rice by increasing ferric reductase activity and the expression of Fe-uptake genes (Aparicio et al., 2023; Sevillano-Caño et al., 2024). Thus, while microbial siderophores may coexist, their overall effect is synergistic rather than antagonistic.
We now explicitly acknowledge in the revised Discussion that microorganisms may also experience Fe limitation and produce their own chelators, which can transiently modify Fe availability in the rhizosphere and, in turn, activate plant Fe-deficiency signaling. This aspect has been incorporated into the revised text to reflect the complexity of microbe–plant interactions under Fe stress.
Round 2
Reviewer 1 Report
Comments and Suggestions for Authors
The authors made significant changes to the text of the article, especially in the methods section, in accordance with the reviewer's comments.
Author Response
Reviewer 1
The authors made significant changes to the text of the article, especially in the methods section, in accordance with the reviewer's comments.
Authors´response: We are deeply grateful for your collaboration. All your comments have been extremely valuable in improving the manuscript and making it more engaging and understandable for the reader. Without a doubt, your dedication and thorough work as a reviewer have added significant value and greatly enhanced the quality of our research team's contribution.

Reviewer 2 Report
Comments and Suggestions for Authors
The author's current experimental results still contain many inconsistencies, making it difficult to draw unified conclusions. The author needs to explain these contradictory aspects before publication.
- For the control group without added microorganisms in Figure 1, Phytosiderophore was significantly induced after 96h under both Fe added and Fe deficient conditions. However, in Figure 2, the Fe added control group was induced at 72h while the Fe deficient control group remained relatively stable. The control group results in these two figures are contradictory.
- The author think that in D. hansenii treatment, Phytosiderophore production is only induced under Fe deficiency conditions. However, in qRT-PCR results, many genes such as OsNAAT, OsTOM1, OsYSL15, and OsIRT1 were induced under both Fe sufficient and Fe-deficient conditions, although there were differences in the induction time.
- Many genes show expression patterns that increase, then decrease, and then increase again. The authors think that these genes exhibit periodic oscillations, but there is no consistent oscillation pattern. For example, in Figure 6c, OsYSL15 is induced at 72h, significantly decreases at 96h, and is then significantly induced again at 120h. In contrast, in Figure 6d, OsYSL15 is significantly induced at both 72h and 96h, but decreases at 120h. Even within the same figure, the oscillation times are not regular; in Figure 8a, the first peak of OsEIN2 appears at 48h, while the second peak appears at 120h. Meanwhile, no oscillation is observed in Figure 8b.
Author Response
Reviewer 2
The author's current experimental results still contain many inconsistencies, making it difficult to draw unified conclusions. The author needs to explain these contradictory aspects before publication.
For the control group without added microorganisms in Figure 1, Phytosiderophore was significantly induced after 96h under both Fe added and Fe deficient conditions. However, in Figure 2, the Fe added control group was induced at 72h while the Fe deficient control group remained relatively stable. The control group results in these two figures are contradictory.
Authors´response: We appreciate your observation regarding the apparent contradiction between the results shown in Figure 1 and Figure 2 for the control groups. The reason for these differences is not fully understood. It should be noted that these were independent experiments conducted with different microorganisms, which may have generated specific interactions that explain the behaviors observed.
Furthermore, as highlighted in another of the reviewer’s comments, plant response mechanisms may be subject to oscillations in the induction of phytosiderophores, resulting in “saw-tooth” patterns. Such fluctuations could account for the temporal variations in induction detected in the controls under both iron-sufficient and iron-deficient conditions.
Finally, we would like to emphasize that the primary objective of our study is to assess the role of microorganisms in phytosiderophore production and in the expression of genes related to their synthesis and transport in rice plants grown with and without Fe. For future publications, it would indeed be valuable to further elucidate the aspects you have kindly suggested.
The author think that in D. hansenii treatment, Phytosiderophore production is only induced under Fe deficiency conditions. However, in qRT-PCR results, many genes such as OsNAAT, OsTOM1, OsYSL15, and OsIRT1 were induced under both Fe sufficient and Fe-deficient conditions, although there were differences in the induction time.
Authors´response: We sincerely appreciate the reviewer’s valuable comment. In the manuscript, the authors have independently discussed the results concerning the effect of D. hansenii on phytosiderophore production, which we consider consistent with the data presented. Based on the methodology employed (Inal et al., 2007), an induction of phytosiderophore production by D. hansenii can indeed be observed. In addition, we have discussed the effect of D. hansenii on the expression of genes related to phytosiderophore biosynthesis (e.g., OsNAAT) as well as those encoding phytosiderophore transporters (OsTOM1, OsYSL15, and OsIRT1).
From our perspective, it is reasonable to state that phytosiderophore production increases in the presence of the yeast under Fe-deficient conditions. However, it should be noted that gene expression does not always correlate directly with the enzymatic activity of the proteins encoded by those genes, since post-transcriptional regulation mechanisms are known to occur (Connolly et al., 2003). These mechanisms can modulate enzyme activity, which may explain the lack of a strict correlation between gene expression and phytosiderophore production. This reason may help explain why the induction of the expression of certain genes related to phytosiderophore production can be observed in plants grown under Fe sufficiency, while no corresponding effect is detected in phytosiderophore production when determined using methodologies such as that of Inal et al. (2007).
Future work by our group may be directed toward elucidating this consideration. We are grateful for the reviewer’s comment, which provides valuable guidance and opens new avenues for further investigation.
Many genes show expression patterns that increase, then decrease, and then increase again. The authors think that these genes exhibit periodic oscillations, but there is no consistent oscillation pattern. For example, in Figure 6c, OsYSL15 is induced at 72h, significantly decreases at 96h, and is then significantly induced again at 120h. In contrast, in Figure 6d, OsYSL15 is significantly induced at both 72h and 96h, but decreases at 120h. Even within the same figure, the oscillation times are not regular; in Figure 8a, the first peak of OsEIN2 appears at 48h, while the second peak appears at 120h. Meanwhile, no oscillation is observed in Figure 8b.
Authors´response: We sincerely acknowledge the reviewer’s observation regarding the oscillations described (Marschner 2012; Lucena et al 2021; Núñez-Cano et al 2023; Aparicio et al 2025). Such fluctuations are indeed well-documented in the literature. It has been established that the response mechanisms activated by plants under Fe deficiency, both in the activity of the enzymes involved and in the expression of the genes encoding those enzymes, do not remain constant over time. Rather, they are subject to cycles of strong induction followed by phases of decline. This dynamic behavior is attributable to the considerable energetic cost required for plants to sustain these responses. To optimize energy expenditure, plants alternate between periods of high induction and periods of low induction, without necessarily following a consistent or homogeneous pattern, as activation and deactivation may also depend on other physiological factors.
Importantly, despite these oscillations, our results consistently show a stronger induction of the gene expression, highlighted by the reviewer, in the presence of the microorganism. This finding supports the central hypothesis of our study: that the presence of the two microorganisms analyzed in this work promotes siderophore production and enhances the expression of genes involved both in their biosynthesis and in Fe transport into the plant. We agree that the oscillatory nature of these responses merits further investigation. This point will be addressed in future research by our group, with the aim of elucidating the underlying causes of the fluctuations observed.

Reviewer 3 Report
Comments and Suggestions for Authors
The authors have considered most of the suggestions and improved the manuscript by additional explantions. Type errors were corrected. However, I still cannot see the value of the indirect estimation of phytosiderophores (PS). As the authors did not intent to quantify absolute DMA concentrations and "previous studies have shown that even Pseudomonas mutants deficient in siderophore synthesis can still enhance Fe acquisition", I recommend to delete this part and to change the text. In this case, no additional references are necessary.
Author Response
Reviewer 3
The authors have considered most of the suggestions and improved the manuscript by additional explantions. Type errors were corrected. However, I still cannot see the value of the indirect estimation of phytosiderophores (PS). As the authors did not intent to quantify absolute DMA concentrations and "previous studies have shown that even Pseudomonas mutants deficient in siderophore synthesis can still enhance Fe acquisition", I recommend to delete this part and to change the text. In this case, no additional references are necessary.
Authors´response: We sincerely appreciate your observation regarding the indirect estimation of phytosiderophores (PS), specifically DMA. While we fully understand your concern, we kindly request permission to retain the information related to DMA in the manuscript. In our view, these data provide valuable insights that complement the main objectives of the study and may be of interest to readers seeking a broader understanding of phytosiderophore production and iron acquisition.
Although our study does not aim to quantify absolute concentrations of DMA, we believe that including these results enriches the discussion and offers a more comprehensive perspective on the mechanisms involved. Therefore, we respectfully suggest maintaining this section, as it contributes meaningfully to the overall scientific value of the manuscript.

Round 3
Reviewer 2 Report
Comments and Suggestions for Authors
For my previous question "The author think that in D. hansenii treatment, Phytosiderophore production is only induced under Fe deficiency conditions. However, in qRT-PCR results, many genes such as OsNAAT, OsTOM1, OsYSL15, and OsIRT1 were induced under both Fe sufficient and Fe-deficient conditions, although there were differences in the induction time", it is recommended that the author further discusses the possible reasons why phytosiderophore production related genes are induced under Fe-sufficient conditions while the phytosiderophore itself decreases in the discussion. If enzyme activity is used to explain the inconsistency in gene expression, then the induction of these genes under Fe-deficient conditions cannot prove an increase in corresponding enzyme activity.
Author Response
Reviewer 2.
For my previous question "The author think that in D. hansenii treatment, Phytosiderophore production is only induced under Fe deficiency conditions. However, in qRT-PCR results, many genes such as OsNAAT, OsTOM1, OsYSL15, and OsIRT1 were induced under both Fe sufficient and Fe-deficient conditions, although there were differences in the induction time", it is recommended that the author further discusses the possible reasons why phytosiderophore production related genes are induced under Fe-sufficient conditions while the phytosiderophore itself decreases in the discussion. If enzyme activity is used to explain the inconsistency in gene expression, then the induction of these genes under Fe-deficient conditions cannot prove an increase in corresponding enzyme activity.
Authors´ response:
The authors sincerely appreciate the reviewer’s valuable comment, which has greatly contributed to improving the manuscript. The requested point has already been incorporated into the Discussion section.
Reviewer 3 Report
Comments and Suggestions for Authors
Dear Authors,
You convinced me. I have no further concerns.
Round 4
Reviewer 2 Report
Comments and Suggestions for Authors
Check the reference list for accuracy and completeness, e.g. ref [11] no author listed.